# Isotope-Assisted Metabolic Flux Analysis: A Powerful Technique to Gain New Insights into the Human Metabolome in Health and Disease

**DOI:** 10.3390/metabo12111066

**Published:** 2022-11-04

**Authors:** Bilal Moiz, Andrew Li, Surya Padmanabhan, Ganesh Sriram, Alisa Morss Clyne

**Affiliations:** 1Department of Bioengineering, University of Maryland, College Park, MD 20742, USA; 2Department of Chemical and Biomolecular Engineering, University of Maryland, College Park, MD 20742, USA

**Keywords:** metabolic flux analysis, fluxomics, isotopic non-stationary metabolic flux analysis, metabolic engineering, metabolomics, mass spectrometry, NMR

## Abstract

Cell metabolism represents the coordinated changes in genes, proteins, and metabolites that occur in health and disease. The metabolic fluxome, which includes both intracellular and extracellular metabolic reaction rates (fluxes), therefore provides a powerful, integrated description of cellular phenotype. However, intracellular fluxes cannot be directly measured. Instead, flux quantification requires sophisticated mathematical and computational analysis of data from isotope labeling experiments. In this review, we describe isotope-assisted metabolic flux analysis (iMFA), a rigorous computational approach to fluxome quantification that integrates metabolic network models and experimental data to generate quantitative metabolic flux maps. We highlight practical considerations for implementing iMFA in mammalian models, as well as iMFA applications in in vitro and in vivo studies of physiology and disease. Finally, we identify promising new frontiers in iMFA which may enable us to fully unlock the potential of iMFA in biomedical research.

## 1. Introduction

Metabolic errors and perturbations play an important role in a variety of human diseases, including cancer [1], neurological disorders [2,3,4,5], and cardiovascular disease [6,7], among others. Although cellular metabolism is often considered simply in terms of energy production and consumption [8], metabolites are tightly linked to critical cellular functions through roles in signal transduction, post-translational protein modification, and gene regulation. In endothelial cells for example, glucose-derived metabolites enter the hexosamine biosynthetic pathway and form O-GlcNAc residues that influence protein post-translational modification and cellular signaling, with important implications for cardiovascular disease [8,9,10,11].

Metabolomic workflows identify and quantify metabolites in these complex intracellular networks [12,13]. Traditional metabolomic approaches identify and measure small metabolite abundances with analytical tools such as mass spectrometry (MS) or nuclear magnetic resonance (NMR). Although these traditional metabolomic approaches focus on quantifying metabolite concentrations, they may not clearly indicate underlying metabolic reaction rates. For example, metabolite accumulation can result from increased upstream pathway reaction rates or decreased downstream reaction rates. Therefore, metabolite concentration measurements alone cannot distinguish between these two states [14,15].

Metabolic reaction rates, also known as metabolic fluxes, more accurately gauge cellular metabolic activity. Overall, the fluxome is constrained by the metabolic inventory of the cell and the associated stoichiometry. Furthermore, the rate of each metabolic reaction depends on several upstream parameters including gene expression and regulation, enzyme concentration, enzyme phosphorylation state, and the concentrations of metabolites associated with the reaction. Therefore, metabolic flux measurements capture the net interplay of the transcriptome, proteome, regulome, and metabolome to offer a comprehensive and dynamic representation of cell metabolic state [16,17,18,19].

Metabolic fluxes can be classified into extracellular fluxes, which cross the cell membrane, and intracellular fluxes, which do not cross the cell membrane. The effects of extracellular fluxes, such as glucose uptake rate or cellular (biomass) growth rate, are observable in the extracellular medium. Therefore, these fluxes can be directly measured by tracking changes in extracellular metabolite concentrations or biomass over time.

Intracellular fluxes cannot be measured directly and must be inferred from their effect on the incorporation pattern of an isotopically labeled nutrient substrate (e.g., ^13^C-glucose) [15] into a biological system (e.g., cell, animal). Metabolic reactions convert the supplied labeled carbon sources to intermediate metabolites and secreted products. The resulting isotopic composition patterns of these metabolites depend on the reaction fluxes and the carbon atom rearrangements [20].

These metabolite labeling patterns can be classified as either isotopologues or isotopomers. Isotopologues differ in mass due to the number of labeled atoms while isotopomers differ in both mass and position. Therefore, an isotopologue can have multiple isotopomers with the same mass but with different positions. While a metabolite with *n* carbon atoms can have up to 2*^n^* isotopomers, only a subset of this gamut of isotopomers can be measured, depending on the analytical tool used. Traditional MS or single-quadrupole MS provides labeling information only on isotopologues, as they cannot distinguish positional differences in isotopomers with the same mass. The range of measurable isotopomers increases with analytical tools that provide atomic position specific information such as NMR and tandem or triple-quadrupole mass spectrometry (MS/MS) [21]. In this review, we refer to instrument-measurable labeling states as isotopomers.

Isotopomer information is typically described in the form of mass distribution vectors (MDVs), which describe each isotopomer’s fractional enrichment (isotopomer abundance normalized to the sum of all isotopomers). MDV isotopomers can range from completely unlabeled isotopomers with zero labeled carbons [M+0] to fully labeled metabolites with all labeled carbons [M+*n*], where *n* is the number of carbons in the metabolite [15,21].

One approach to analyze isotope labeling data is to manually assess the MDVs, which provide qualitative insight into differential metabolic activity among experimental groups [22]. The concept is intuitive in simple metabolic systems (Figure 1A), as increased metabolic activity will lead to increased metabolite labeling enrichment in the given pathway. However, interpreting MDV labeling patterns can become difficult in real-world metabolic networks due to factors such as natural isotope abundance, compartmentalization, reversible/cyclic reactions, and complex atom rearrangements [20,23]. For example, an isotope-labeled metabolite generates a unique set of isotopomers as it is incorporated into the first, second, and third rounds of the tricarboxylic acid (TCA) cycle (Figure 1B). TCA MDV complexity scales with the number of carbon atoms, since molecules with *n* carbon atoms can have 2*^n^* MDV isotopomers. Thus, large-scale complex metabolic networks are difficult to analyze manually [15,20].

Isotope-assisted metabolic flux analysis (iMFA) is a powerful computational approach for accurately determining the fluxome from isotope labeling data and a metabolic network model. We use the term iMFA throughout this paper, although MFA and ^13^C-MFA are also sometimes used, to differentiate the approach from flux balance or flux variability analysis and to broaden the scope beyond ^13^C to also include ^15^N and ^2^H labels. iMFA is a complex procedure with many nontrivial steps, including choosing an appropriately labeled carbon source, making the appropriate measurements, and developing a comprehensive isotopomer balance model. iMFA also generally requires the system to be at metabolic (no net metabolite loss or accumulation) isotopic (constant isotope enrichment) steady state and assumes a homogenous cell type in a uniform microenvironment. However, despite these challenges and limitations, iMFA enables analysis of reversible, cyclic, and highly connected metabolic pathways; natural isotope abundance and complex atom rearrangements; metabolite compartmentalization inside the cell; and parallel labeling experiments with multiple isotope labeled tracers.

In this review, we discuss iMFA in detail, including underlying concepts, practical considerations, and examples of how it has been deployed to unveil new insights into human health and physiology.

## 2. Isotope-Assisted Metabolic Flux Analysis (iMFA)

Isotope-assisted metabolic flux analysis (iMFA) uses a combination of experimental external flux and isotope labeling data, metabolic network models, and computation to indirectly quantify intracellular metabolic fluxes that cannot otherwise be measured [20]. iMFA quantifies fluxes by exploiting the relationship between flux values and MDV labeling patterns, typically under metabolic steady-state conditions. The metabolic steady-state assumption stipulates that there is no net metabolite accumulation or consumption. Therefore, production fluxes must equal consumption fluxes. In metabolic steady-state, each metabolite’s MDV depends on the MDVs of its precursor substrates, with the contribution from each precursor weighted by its reaction flux magnitude [24]. As the magnitude of the metabolite production reaction fluxes change, so do the resulting product MDVs.

Figure 2A displays a simplistic metabolic model with two uptake fluxes (*V1* and *V9*), three secretion fluxes (*V7*, *V8*, and *V10*), and five irreversible forward reaction fluxes (*V2*, *V3*, *V4*, *V5*, *V6*). Metabolite B’s MDV is entirely dependent on the MDV of the labeled nutrient tracer A that is fed into the system. However, metabolite F has multiple sources, and its labeling pattern depends on the flux-weighted contributions of metabolites C, D, and externally sourced F. In the scenario in which *V1* and *V10* are the only non-zero extracellular fluxes (Examples 1 and 2), F’s MDV is entirely dependent on the flux values of *V4* and *V5*, which must equal *V2* and *V3*, respectively, (Figure 2B). The [M+1]/[M+2] ratio of F directly informs the percentage of B that is metabolized through reaction *V2* versus *V3*. The absolute flux values of *V2* and *V3* can then be calculated by multiplying the absolute value of the incoming flux *V1* by the proportional values of *V2* and *V3* determined from MDV analysis. The range of absolute flux values is constrained by incorporating additional extracellular flux values, which establish boundaries that cannot be violated. In this example, external F is 100% unlabeled and its intake dilutes the labeled F pool. In Example 3, one-third of the metabolite F pool is sourced from extracellular unlabeled F through *V9*, which limits the fraction of labeled F (Figure 2C). If reactions *V7* and *V8* are also active as in Examples 4 and 5 (Figure 2D), a portion of the labeled C and D are secreted before they can be metabolized to F, further diluting the labeled F isotopomers. Although the flux values and corresponding isotopomer distributions in this example can be calculated by hand, real metabolic systems are much larger and include reversible and cyclic pathways, which make manual tracer analysis infeasible. 

iMFA addresses the complexity of this problem by mathematically formulating the relationship between MDVs and fluxes into a set of mass balance equations that relate metabolic fluxes and isotopomers [25]. This leads to the mathematical problem of determining a set of flux values that best accounts for the isotopomer data. This parameter estimation problem is best solved by optimization. In the traditional iMFA used by all MFA software prior to 2021, the optimization begins with an initial guess for all metabolic fluxes in the system. These fluxes are then used to generate an initial simulated MDV for each metabolite in the model. The computationally simulated MDVs are compared to the experimentally measured MDVs. The iMFA model then iterates this process [23] by optimizing estimated fluxes to minimize the difference between the simulated and experimental MDVs [20,24]. Ultimately, the error between experimental and simulated MDVs is minimized to a statistically acceptable fit [26]. In a recently reported approach, the iMFA problem is solved by using state-of-the-art optimization tools on an algebraic modeling tools such as the General Algebraic Modeling System (GAMS; https://www.gams.com). This enables the labeling state to be updated instead of being calculated explicitly from scratch during each iteration of such an optimization, providing for greater robustness and scalability to large networks [27]. Finally, iMFA generates a quantitative flux map of the entire metabolic network, including statistical confidence intervals for each flux value [25]. A detailed discussion of the mathematics underlying iMFA is beyond the scope of this article and can be found elsewhere [24].

While other techniques such as manual tracer analysis, isotopomer spectral analysis (ISA), flux balance analysis and sampling, flux ratio analysis, and local mathematical models) can be used to calculate metabolic fluxes, iMFA offers key advantages over manual tracer analysis as well as local, focused mathematical approaches [28,29,30,31]:
iMFA provides quantitative information on the entire metabolic network, including metabolic flux values, confidence intervals, and statistical analysis.The iMFA model can easily be modified with new pathways, compartments, or influxes, since the iMFA software automatically reformulates the underlying metabolic network when a reaction is added or removed [32]. This feature is especially valuable when the preformulated metabolic network does not satisfactorily fit the data and can be advantageous over other flux analysis methods that require analytical formula derivations for each flux [6,33].A poor fit between the iMFA model and labeling data suggests either measurement errors or incorrect model assumptions. Modifying the metabolic model to achieve an acceptable fit can uncover previously unknown metabolic features and may ultimately lead to new insights into the metabolic system, such as unconventional metabolite channeling [34] or novel major carbon sources [35].iMFA rigorously accounts for network complexities [20,23], including: reaction reversibilities, which are common in the pentose phosphate pathway (PPP) and the tricarboxylic acid (TCA) cycle; pathway cyclicity (a TCA cycle feature); high network connectivity (common in central carbon metabolism); and isotope natural abundance.iMFA is particularly useful for discerning fluxes in complex mammalian cells that have multiple inputs, which can complicate interpretation of labeling patterns. For example, in glucose labeling experiments, the labeling of TCA metabolites is diluted by anaplerotic compounds. By integrating isotope labeling data with extracellular flux values, iMFA can address whether decreased labeling is due to decreased entry of labeled nutrients, increased incorporation of unlabeled compounds, or both.iMFA is scalable and can incorporate large metabolic data sets. In addition, the network-based approach of iMFA ensures that labeling patterns are analyzed in the context of the whole network, rather than as standalone elements. These attributes allow for the iMFA approach to fill holes in the model and buffer measurement error effects in the data [6,32].Although iMFA and ISA are closely related and can often be performed using the same software, ISA is typically used to estimate fractional contributions of different metabolites to de novo fatty acid biosynthesis, whereas iMFA estimates absolute metabolic fluxes throughout the metabolic network [20,36,37].While in silico genome-scale model flux analysis is a powerful fluxomic tool, it relies on numerous assumptions for flux prediction. iMFA estimates metabolic fluxes from actual isotope labeling experiment data and offers superior flux resolution [38].

### iMFA Workflow

Three inputs are required for standard iMFA: experimental external fluxes, experimental MDVs, and a metabolic map that delineates all reactions of interest (Figure 3) [20]. External fluxes, including nutrient intake (e.g., glucose, glutamine) and metabolic byproduct secretion (e.g., lactate, glutamate), constrain the metabolites that enter and leave the intracellular metabolic network. To determine the external fluxes in vitro, extracellular metabolite concentrations can be measured by high-performance liquid chromatography (HPLC) [39,40,41,42], gas/liquid chromatography (GC/LC)-mass spectrometry (MS) [43,44], enzymatic assays, or YSI biochemistry analyzers [35]. Changes in extracellular metabolite concentrations are then converted into external fluxes normalized by time, cell number, and the cell growth rate. Confounding factors, such as nutrient and metabolic byproduct degradation, as well as liquid evaporation, should be accounted for when calculating external fluxes [20]. Extracellular fluxes are more complicated to measure in vivo due to the complex metabolite exchange among tissues. For these studies, measuring labeled metabolite levels in blood extracted from varied vessels is essential to determining metabolite consumption and secretion from different organs [45,46]. External biomass flux can also be calculated from experimentally measured cell or tissue biomass and incorporated in the model by defining how each metabolite contributes to overall biomass production.

Experimental MDVs are acquired by feeding isotope-labeled tracer nutrients to cells or tissues. Labeled carbon tracers are commonly used to study the major metabolic pathways. For example, [U-^13^C] glucose provides information on glucose contribution to the TCA cycle, while labeled glutamine provides information on α-ketoglutarate reductive carboxylation and gluconeogenesis [47,48]. Non-carbon tracers can also be used. For example, deuterium (^2^H) is used to investigate mitochondrial and cytosolic NADPH metabolism [49]. After the biological system has been fed a specific amount of a given tracer nutrient for the desired time, the cells or tissue are quenched and metabolites are extracted for analysis. Metabolites are then analyzed by mass spectrometry (MS) or nuclear magnetic resonance imaging (NMR), resulting in a dataset for each metabolite of interest and its corresponding MDV. 

The iMFA metabolic map is a table of metabolites, reactions, and their corresponding stoichiometry. Metabolic maps range in complexity, but due to experimental limitations, most mammalian cell maps focus primarily on glycolysis and the TCA cycle. iMFA metabolic maps are constructed from the literature and metabolic databases such as Kyoto Encyclopedia of Genes and Genomes (KEGG) [50], Virtual Metabolic Human [51], and MetaCyc [52]. Reactions are classified as source, sink, or internal reactions. Source reactions involve metabolites that are consumed but not produced by another network reaction. Examples include glucose uptake from the extracellular compartment or acetyl-CoA derived from intracellular fatty acid oxidation. Sink reactions involve metabolites that are produced but not consumed by other reactions. These reactions can be used to represent either metabolite efflux (such as lactate secretion) or intracellular storage (such as glucose fated for glycogen synthesis or acetyl-CoA routed for fatty acid synthesis). Finally, internal reactions involve metabolites that are both consumed and produced by other network reactions. Internal reactions typically are intracellular reactions that link source and sink reactions, such as the glycolytic reactions that derive lactate from glucose. Source, sink, and internal reactions are then organized into the compartments (e.g., cytosol, mitochondria) where each reaction takes place.

Once the reactions are specified in the iMFA metabolic map, atom transitions must be defined for each biochemical reaction to allow the iMFA model to track labeled atom movement. Atom transition information can be attained from the literature as well as the previously mentioned KEGG, Virtual Metabolic Human, and MetaCyc databases [48,49,51]. However, there are still limitations in the comprehensiveness and accuracy of available atom-mapping information. Efforts to harmonize atom-mapping information among different databases will improve the accuracy and consistency of available atom-transition information [52]. There are also efforts to develop novel algorithms to map atom transitions in complex organic reactions and ultimately improve biochemical pathways [53]. 

The external fluxes, MDVs, and metabolic maps are then integrated using software such as eiFlux [27], INCA [21,54], METRAN [55], OpenMebius [56], and 13C2FLUX [57], which support model construction, flux estimation, and confidence interval determination. In most iMFA software, the user enters a list of reactions of interest along with their associated metabolites and stoichiometric balances to define the metabolic network. User-entered experimental external flux and MDV data is mapped to the corresponding reactions or metabolites in the model. The MFA software then begins an iterative process that determines the best set of internal fluxes that fit with the external fluxes and MDVs, given the entered metabolic model. Sometimes the software does not achieve a satisfactory fit due to (i) data errors or (ii) incorrect model assumptions [58]. The latter scenario is often addressed by carefully comparing experimental MDVs with simulated MDVs and reformulating the model to include, exclude, or modify reactions. Visualization tools such as Escher-Trace [59] are useful in visualizing multiple MDVs simultaneously to determine where a compartmentalization reaction or additional metabolite may be needed to fit the experimental data. Once a satisfactory fit is achieved, the model outputs flux estimates for each network reaction. The uncertainty in the flux map can be estimated by determining confidence intervals of the fluxes. Currently, two methods are used for this purpose. In the bootstrap method [23], the measured isotopomer dataset is perturbed to create numerous synthetic isotopomer datasets that are distributed (typically in a normal distribution) around the measured dataset. Fluxes are evaluated from each of the synthetic datasets, and the distribution of these fluxes is used to determine the flux confidence intervals. In another method suited to iMFA [60], the confidence intervals are found by determining a set of fluxes that accounts for the measured isotopomer dataset and then varying each flux stepwise until it reaches a χ2 threshold or its stoichiometric limit [24]. The model output can be formatted into visual flux maps using semi-automatic visualization software such as FluxVisualizer [61].

## 3. iMFA Considerations for Mammalian Cells

### 3.1. Tracer Selection

Labeled tracer nutrients should be strategically chosen to maximize information for the experimental objective. In general, there are three important tracer nutrient parameters to consider when choosing a labeling scheme: the specific labeled tracer nutrients used in the study, their relative abundance (e.g., the proportion of the metabolic substrate that the labeled tracer nutrient constitutes), and tracer mixes/parallelization. 

The tracer nutrient and the positions of its labeled atoms partially determines the MDV labeling pattern. While uniformly labeled tracer nutrients provide information on major metabolic pathways at a high level, tracer nutrients with a specific number and location of labeled atoms can elucidate complexities at specific branches. For example, the first carbon of glucose-6-phosphate is lost as carbon dioxide (CO_2_) at the branch point from glycolysis to the pentose phosphate pathway (PPP). [1,2-^13^C] glucose is ideal for measuring this branch point, as glucose produces M+2 lactate when it is metabolized through glycolysis and M+1 lactate as it is metabolized through the PPP [62]. In contrast, uniformly labeled glucose is metabolized into M+3 lactate in both scenarios and thus cannot be used to distinguish these two pathways (Figure 4). Detailed tables explaining which tracers are ideal for each metabolic pathway are found elsewhere [14]. 

Tracer nutrient abundance refers to how much of the nutrient is labeled with the tracer relative to unlabeled nutrient. Similar to the choice of labeled tracer, nutrient abundances can also impact the labeling patterns that are generated. For example, uniformly labeled glucose is most informative when mixed in a 1:1 ratio with unlabeled glucose [63]. Feeding cells 100% [U-^13^C] does not provide information on the PPP branchpoint, while using a 50% [U-^13^C] glucose and 50% unlabeled glucose scheme will generate F6P and G3P isotopomers that are only generated in the PPP. In contrast, mixing with the unlabeled tracer form reduces the effectiveness of any tracer that contains both ^13^C and ^12^C atoms in the same molecule (e.g., 1-^13^C glucose or 1,2-^13^C glucose) [63].

For this reason, isotope labeling experiments should be carefully designed to use the optimal tracer substrates, abundance, and combinations to optimally achieve experimental goals. To address these concerns, computational and algorithmic approaches have been developed to optimize tracer selection and abundance in mammalian cells, since performing tracer experiments can be expensive and time-consuming. In an early example, sensitivity analysis was used to determine that 100% [1-^13^C] glucose is the most precise scheme when only considering PPP and glycolysis in rat hepatoma cells, while 100% [U-^13^C] glucose offer superior precision for TCA cycle and anaplerotic pathways. An approximate 51:44:5 ratio of [1-^13^C]/[U-^13^C]/unlabeled glucose was used to optimize precision for all pathways [64]. Later studies expanded on this work to assess precision in mammalian systems fed with multiple substrates. The authors developed a flux precision scoring algorithm to identify optimal tracers for specific mammalian cell pathways. This algorithm calculates the magnitude of the flux confidence interval for each flux in a system, converts it into a precision score by using a negative exponential function, and computes the overall weighted-averaged precision score. The overall score is then compared between two tracer schemes to assess differences in precision. Using this method, [1,2-^13^C] glucose was determined to provide the most precise estimates for glycolysis, PPP, and overall metabolism, while [U-^13^C] glutamine was better suited for the TCA cycle [65]. 

A follow up study used precision scores to analyze a scheme where glutamine and glucose labels were simultaneously fed to cell cultures in one experiment, rather than in separate experiments as occurs in parallel labeling approaches. The algorithm simulated measurements for each tracer mixture, estimated fluxes, calculated intervals, and selected the most precise combinations. Using this approach, [1,2-^13^C] glucose/[U^13^C_6_] glucose and [3-^13^C] glucose/[3,4-^13^C] glucose/[U^13^C_5_] glutamine were identified as optimal, non-redundant tracer mixtures to study glycolysis and TCA cycle in non-small cell lung cancer cells [66]. 

Parallel labeling experiments, in which isotope labeling experiments are repeated with different labeled tracers targeted at specific pathways and then integrated into one iMFA model, can also be used to maximize estimated iMFA flux precision [66]. Each labeled tracer provides complementary information, that can be integrated into a single iMFA model [6,20]. For example, labeled glucose can be used to evaluate glycolysis and the PPP, while in parallel labeled glutamine can be used to evaluate reductive carboxylation, TCA, and lipogenesis [64,67]. A recent scoring system to identify optimal parallel labeling schemes has also been described, although this study only tested and validated results in E. coli [68]. A parallel labeling scheme of [1,6-^13^C] glucose and [1,2-^13^C] glucose increased flux precision by 18-fold when compared to the reference labeling scheme of 81% [1-^13^C] glucose. In comparison, non-parallel individual labeling only achieved approximately 5-fold precision score increases [68]. The mathematical concepts underlying tracer selection are beyond the scope of this review and can be found in the articles referred to in this section [63,65,66,68].

Tracer selection is made more accessible to novices and experts alike through the inclusion of user-friendly versions in MFA software packages. For example, INCA offers a tracer simulation platform, which provides steady-state or transient MDV simulations based on the selected nutrient tracer and user-estimated fluxes [54].

### 3.2. Steady State Considerations

Metabolic steady state occurs when metabolic parameters such as nutrient uptake and cell growth are constant over time. There is therefore no net metabolite accumulation in the system as mass influx and efflux are balanced. Thus, although each flux has a particular value, the net sum of the fluxes must equal zero at steady state. Cell cultures are assumed to be at metabolic steady state when they are either not proliferating or when the growth rate is constant (e.g., during the exponential growth phase). The metabolic steady state assumption eliminates kinetic parameters [69] and simplifies the iMFA mathematical problem from a series of ordinary differential equations into a system of algebraic equations. Practically, iMFA experiments are usually conducted in cells at pseudo metabolic steady state, defined by stable metabolic parameters over the experimental timescale [15]. Both exponentially growing and non-proliferating cells are assumed to be at pseudo-metabolic steady state [15].

Isotopic steady state occurs when isotope enrichment remains constant over the experimental timescale [15]. The time required to reach isotopic steady state varies across different cell types, metabolic pathways, and tracers. However, isotopic steady state can be difficult to achieve in mammalian cells due to high exchange between intracellular and extracellular metabolite pools, particularly amino acids, leading to slow labeling [22,70]. Furthermore, it is difficult to maintain metabolic steady state long enough for the system to reach isotopic steady state [22,71] Achieving isotopic steady state is further complicated under conditions such as pharmacological cell treatment or growth factor withdrawal [22]. When isotopic steady state cannot be reached, metabolic pathway labeling dynamics can be analyzed by isotopic non-stationary MFA (INST-MFA) [14]. INST-MFA still requires the system to be at metabolic steady state, but total intracellular metabolite pool size [72] is additionally used to estimate isotopomer distributions over time [25]. This has the added benefit of allowing estimation of intracellular metabolite concentrations [71]. INST-MFA is a more computationally demanding approach since the isotopomer balances are ordinary differential equations. However, many of the previously listed software such as INCA, eiFlux, and OpenMebius can perform INST-MFA [54,56,73].

### 3.3. Quenching

Quenching terminates cellular metabolic activities so that metabolites can be extracted from cells or tissues at the desired state. Efficient quenching is crucial, as metabolites quickly degrade during sample handling, samples can be contaminated with extracellular metabolites in the supernatant, and enzymatic activity can change intracellular metabolite levels. Typical quenching protocols for adherent cells involve aspirating media, washing with phosphate-buffered saline (PBS), and then immediately applying a quenching solvent. In contrast, suspension cell cultures must be quickly filtered to remove the media and then placed in the quenching solvent [74]. Enzyme activity can be halted by adding hot or cold organic solvent. Hot solvents effectively denature enzymes but can lead to thermal degradation, while cold solvents avoid degradation but at the cost of not fully denaturing enzymes and potentially allowing metabolic reactions to continue. Continued enzymatic activity or metabolic degradation can be halted by adding acidic and basic solvents following quenching. Typically, mammalian samples are quenched using a cold acidic organic solvent, such as 80:20 methanol:water or 40:40:20 acetonitrile:methanol:water. Typically, both are sufficient to extract polar metabolites, although the solvent composition may be fine-tuned to optimize extraction of compounds such as ATP and NADPH. Following extraction, samples are dried (via lyophilization, speed vac systems, or under nitrogen gas) and processed for further analysis.

### 3.4. Measuring MDVs: MS vs. NMR

The three principal techniques to experimentally measure MDVs are liquid chromatography coupled to mass spectrometry (LC-MS), gas chromatography coupled to mass spectrometry (GC-MS), and nuclear magnetic resonance (NMR) [75]. Mass spectrometric methods ionize molecules, separate them by the mass-to-charge ratio (*m*/*z*), and measure them with a detector. Mass spectrometers are typically paired with chromatographic methods to further resolve spectra and separate isomers, compounds with identical molecular formulas and *m*/*z* ratios but different structures [74,76]. In contrast, NMR applies strong magnetic fields and radio frequency pulses to atoms, causing their nuclei to spin and emit energy upon relaxation. This energy is measured in the NMR spectrum, which can be used to identify and quantify metabolites. Generally, mass spectrometry is more sensitive and allows detection of sparsely abundant isotopomers but lacks the specific structural information provided in NMR. 

Mass spectrometry is highly sensitive, allowing thousands of compounds to be detected at micromolar or nanomolar concentrations [75,77]. This sensitivity reduces the sample size needed for analysis, which can be important for high throughput cell culture or when taking multiple samples from an animal model [32]. Although mass spectrometry methods can detect low metabolite concentrations, they are limited in distinguishing isotopomers from natural abundance background when isotope enrichment is low [78].

LC-MS and GC-MS are the two primary separation techniques paired with mass spectrometry. LC-MS is a highly versatile method that uses mobile solvents (such as acetonitrile, water, and methanol) to pass metabolites through a column and separate them by properties such as polarity, charge, and size. Following separation, metabolites are ionized through electrospray ionization or atmospheric pressure ionization [76]. Sample preparation is simpler in LC-MS because, in contrast to GC-MS, metabolites do not typically have to be derivatized, although they can be derivatized to improve signal or ionize compounds that are resistant to ionization [76,79]. The other primary advantage of LC-MS is that it can detect a broader range of molecules, including low-abundance phosphorylated metabolites and cofactors [77,79]. Despite its broad coverage, there is currently no single LC method that can simultaneously measure polar and nonpolar compounds. Reverse-phase chromatography separates nonpolar compounds such as fatty acids but cannot retain polar compounds, while hydrophilic-interaction chromatography separates polar compounds. Furthermore, LC-MS is prone to ion suppression effects in which high-abundance ions reduce ionization and signal of coeluting ions [74,79]. For most iMFA applications, these drawbacks are minimized by the fact most current metabolites of interests are polar, but this limitation may need to be overcome for iMFA encompassing nonpolar metabolites. 

GC-MS enables efficient metabolite separation but requires chemical derivatization to increase volatility for improved detection, which complicates sample preparation. However, GC-MS allows superior detection of low-molecular weight and volatile species, improved metabolite separation, and reduced ion suppression effects [77]. GC-MS instruments are also less expensive to set up and maintain compared to LC-MS instruments, and GC-MS is preferred over LC-MS for compounds such as sterols, sugars, and very-short-chain fatty acids [74,80]. GC-MS is less laborious and requires smaller samples than LC-MS, but LC-MS can detect a larger range of metabolites [77]. Indeed, some compounds cannot be detected via GC-MS, which means that proxy metabolites or closely related intermediates must be used instead. For example, GC-MS cannot directly detect PPP intermediates (e.g., ribulose-5-phosphate) and instead relies on distal components derived from the PPP (e.g., the amino acid histidine, whose carbon backbone is derived from ribulose-5-phosphate). Despite the limited coverage of GC-MS compared to LC-MS, it provides sufficient information on the metabolites typically used in iMFA [80]. 

NMR requires minimal sample preparation and does not require chromatographic methods to separate metabolites since the spectra provide sufficient structural information for compound identification [75]. NMR can also detect position-specific labeling [78]. For example, NMR has been used to identify labeling in the second and third carbon positions of lactate following incubation with [2-^13^C] glucose, allowing discrimination of PPP vs. glycolytic flux. This aspect of NMR is especially helpful to determine which atoms in a metabolite are labeled, especially when multiple tracers are used simultaneously [78]. A major disadvantage of NMR is its lack of sensitivity, which means that a larger sample is needed for detection [74,75,76]. For these reasons, MS is typically better suited for cell culture or animal samples in which metabolite concentrations fall in the micromolar range.

Tandem mass-spectrometry (Tandem MS) is an emerging technique with the benefits of both MS and NMR. Tandem MS uses collision energy to break precursor ions into smaller product ion fragments. This fragmentation allows position specific atom information to be derived from MS data [81]. Tandem MS has increased accuracy when compared to traditional GC-MS and LC-MS and also provides broader coverage by improving detection of low abundance isotopomers [82]. Tandem MS is challenging to use as an iMFA input because the mathematical iMFA framework must be expanded to integrate the fragmented isotopomer information. Algorithms were recently described but are not commonly implemented in existing MFA software [83]. To the best of our knowledge, eiFlux is the only MFA software that allows users to use Tandem MS data [27].

### 3.5. Natural Isotope Abundance

Natural isotope abundance corrections must be conducted when quantifying MDVs or incorrect conclusions may be drawn. Natural isotope abundance describes the heavy hydrogen, oxygen, sulfur, and nitrogen isotopes that naturally occur in biological metabolites and sample reagents. For example, in an experiment that uses a ^13^C tracer, naturally occurring ^13^C, ^15^N, and ^18^O in the metabolite may increase the measured molecular mass and cause the metabolite to be classified as a heavier isotopomer despite the lack of heavy carbon incorporation. Matrix-based correction can be used to correct for natural isotope abundances before the data are entered into the iMFA software. A detailed description is beyond the scope of this review and can be found elsewhere [84,85]. Some software suites, such as INCA, can automatically correct for natural abundance isotopes. However, as tandem MS data is incorporated into iMFA, these algorithms will need to be updated to perform more complex corrections.

### 3.6. Compartmentalization

Compartmentalization refers to specifying in the iMFA model that metabolites and metabolic reactions are in specific cellular compartments (e.g., mitochondria, peroxisome, lysosome), just as they are in the biological system. However, since quenching mixes metabolites from all cellular compartments, experimental MDVs reflect pooled metabolites from the whole cell [20]. Compartmentalization must be included to enable a good fit between simulated and experimental MDVs in the iMFA model. For example, pyruvate is in both the mitochondrial and cytosolic compartments. Alanine is produced from mitochondrial pyruvate, while lactate is produced from cytosolic pyruvate [86]. A unicompartmental model will assume that both alanine and lactate have nearly identical MDVs to the pooled pyruvate, since there is no carbon rearrangement when interconverting between these molecules. This may ultimately lead to a poor fit or biased flux estimates [87]. In a non-stationary iMFA model of Chinese hamster ovary cells [70], the non-compartmentalized model failed to converge and resulted in a poor fit due to low labeled pyruvate, alanine, lactate, and glutamate levels. A second model was created with a mitochondrial compartment separate from the cytosol. The updated model successfully fit the data and predicted that malic enzyme, which catalyzes malate conversion to pyruvate, was active only in the mitochondria. These results demonstrate the importance of compartmentalization, especially as reactions and metabolites are often sequestered in mammalian models. Compartmentalization can be accounted for in the iMFA software [54] or experimentally. For example, INCA enables the user to declare all compartments in which a metabolite may exist, predicts the relative contribution of each compartment to the overall metabolite pool, and scales the simulated MDV accordingly.

Experimental approaches such as reporter metabolites [88,89,90], fractionation and organelle isolation [91,92,93,94], gene expression measurements [87] digitonin permeabilization [95], and protein localization/GFP-fusion-type localization/computational inference transit peptides can be used to infer compartmentalization effects. Organelle isolation in particular can directly quantify compartmental metabolites; however, flux studies based on purified or isolated organelles may not be as physiologically relevant as those in a whole cell [87,96]. In addition, organelle isolation can be a lengthy process that can perturb metabolism. Recently, a spatial-fluxomics approach was developed to achieve subcellular fractionation and cell quenching within 25 seconds. Fractions remained relatively pure (~90%) with less than 20% loss of metabolite pool sizes when compared to whole-cell methods [96]. Through this technique, the authors discovered that succinate dehydrogenase deficient tumors do not export citrate to the cytosol and instead reverse citrate synthase flux to produce mitochondrial oxaloacetate for pyrimidine biosynthesis.

### 3.7. Dilution Reactions

Dilution reactions are included in iMFA models to account for unlabeled carbon sources that enter the metabolic network and decrease the metabolite labeled fraction. A key sign of dilution is a high abundance of an [M+0] label in metabolites that should otherwise be fully labeled, such as in G6P in cells provided with [U-^13^C] glucose. Potential dilution sources include additional carbon influxes (e.g., amino acid uptake) and intracellular carbon use (e.g., glycogen breakdown). In these cases, dilution sources provide unlabeled carbons that propagate throughout the network and contribute to net metabolite flux. In other cases, dilution sources carry no net carbon flux since there is an equal rate of breakdown and synthesis, such as when there is an exchange of labeled and unlabeled amino acids that dilutes the labeled amino acid pool but does not change net carbon flow. 

Compartmentalization and metabolite channeling can also lead to dilution as physical and enzymatic separation prevents labeled atom incorporation into specific metabolite pools, leading to distinct labeled and unlabeled populations of the same metabolite. Channeling is the occurrence of multiple, consecutive reactions on the same enzyme complex without the intermediates equilibrating in the cell or compartment [97]. This can lead to unexpected isotopomers distributions as metabolite mixing is bypassed. For example, lactate is derived from pyruvate, which itself can be derived from carbon sources such as glucose or amino acid. If metabolite channeling occurs between the enzymes that metabolize amino acids into pyruvate and subsequently lactate, this will bypass glucose contributions to pyruvate, leading to a greater portion of unlabeled lactate than expected in a glucose tracer experiment [98]. 

By examining where the dilution occurs and whether it propagates to downstream networks, users can identify and test the most likely causes of dilution. Possible network modifications include adding novel dilution fluxes, compartmentalization, or pathway bypasses [35,98]. This can help identify unconventional pathways, as shown in our previous work [35]. 

### 3.8. Tissue-Specific Model

Tissue-specific models consider the specific metabolism and function of each cell type when constructing the iMFA metabolic map. For example, gluconeogenic pathways are important in hepatocytes [99] and should be included in hepatocyte iMFA models. In contrast, the core metabolic model should be pruned when studying red blood cells, which lack mitochondria and therefore do not use the TCA cycle [100]. Instead, the red blood cell iMFA metabolic map should focus on pathways essential to red blood cell metabolism, such as glycolysis, PPP, adenosine nucleotide metabolism, and the Luebering-Rapoport shunt pathway [101].

### 3.9. In Vivo iMFA Considerations 

In vivo iMFA provides valuable insights into metabolism in the entire organism. However, due to complex animal physiology, additional considerations must be considered in vivo. Just as with in vitro studies, the in vivo study tracer must be chosen to specifically represent flux through the tissue of interest, with the additional consideration that it must not be significantly influenced by other tissues. Metabolite labeling patterns can be compared in perfused tissue ex vivo and in the animal in vivo to determine the ideal tracer for a specific tissue [102]. There are also tradeoffs when deciding between MS and NMR for in vivo MFA. Since MS is highly sensitive, smaller sample sizes are needed, which minimizes disturbance to the animal especially during repeat sampling. In contrast, NMR is useful for detecting position-specific labeling with multiple tracers, but low NMR sensitivity requires collecting larger samples which can induce stress in small animals [32]. Thus, similar to in vitro iMFA, MS or NMR should be selected based on the required measurements.

Typically, in vivo iMFA experiments start with catherization surgery that allows for continuous infusion of stable isotope tracers. Animals are usually provided both a bolus and continuous infusion of stable isotope tracer, allowing metabolites to reach isotopic steady state. Acute interventions, such as exercise or insulin stimulation, are often conducted following isotope infusion. Following the labeling and experimental steps, tissue and blood samples are collected, analyzed, and integrated into an iMFA model (Figure 5).

Most in vivo flux studies thus far focused on hepatic gluconeogenesis, with labels carefully selected to maximize the information on hepatic gluconeogenesis that can be obtained from in vivo models. Examples of these labels include deuterium oxide (D_2_O), [U-^13^C] propionate, and a labeled glucose tracer (e.g., [6,6-^2^H_2_]-glucose or [3,4-^13^C_2_] glucose) [32,102,103]. Deuterium oxide provides labeled hydrogen atoms that are incorporated into gluconeogenic precursors derived from glycogen, glycerol, and TCA intermediates. The resulting hydrogen enrichment patterns in plasma glucose are used to delineate the relative contribution of these pathways to hepatic glucose production [104,105]. [U-^13^C] propionate is metabolized into succinyl-CoA and subsequentially oxaloacetate but avoids oxidative pathways. [U-^13^C] propionate derivatives then pass through phosphoenolpyruvate carboxylase, gluconeogenesis, pyruvate cycling, and the TCA cycle. Together, propionate and deuterium labeling can be integrated to understand the relationship between TCA cycle fluxes and glucose production [106]. Labeled glucose tracers typically act as an “external flux” measurement. As endogenous glucose is released into the blood, it dilutes the infused labeled glucose. The dilution rate is then used to estimate the gluconeogenic rate and subsequentially to convert relative fluxes to absolute fluxes [78]. 

In vivo models require the assumption that enriched labeled isotopes equilibrate in the tissue and have negligible effects on in vivo fluxes even when administered in levels required for sufficient label enrichment. However, these assumptions should be tested in each in vivo case. In one example, a base hepatic model assumed full TCA intermediate equilibrium and no carbon dioxide or secondary tracer reentry. The model was then tested with ^13^C_3_ propionate/^2^H and ^13^C_3_ Lac/^2^H tracers. Pyruvate cycling was higher in the iMFA model using ^13^C_3_ propionate/^2^H, showing that the labeling scheme affected model predictions. The base model was then expanded to account for Cori cycle fluxes that traffic metabolites between hepatic and extrahepatic compartments and no longer assumed equilibration in certain reactions [107]. The liver pyruvate cycle flux estimates from the updated model were then similar between the two tracers. This demonstrates the importance of carefully testing model assumptions to ensure they are valid for the system of interest. 

## 4. iMFA Applications in Human Physiology and Disease

Many studies harness iMFA to study microbes and algae for industrial applications, including identifying unique metabolic features and potential production bottlenecks. However, in the past decade, iMFA has increasingly been applied to study human metabolism in both physiological as well as pathological states. Here, we describe selected examples from the literature, with an emphasis on how iMFA uncovered new findings as compared to MDV labeling pattern analysis alone. 

### 4.1. Stem Cell Differentiation and Proliferation

Cellular metabolism is closely intertwined with stem cell pluripotency and differentiation [108,109]. Recent studies of differentiating stem cells employed iMFA to elucidate the metabolic needs and rearrangements that emerge during differentiation. In one example, Sá et al. used INST-MFA to understand how metabolism changes as neural stem cells (NSCs) differentiate into astrocytes. Extracellular flux measurements showed that astrocytes consumed less glucose yet produced more lactate than NSCs, and that astrocytes secreted glutamine while NSCs consumed it. Using INST-MFA, the authors discovered that NSCs metabolized glutamine to citrate by reversing α-ketoglutarate flux, which could support the lipogenesis important for NSC proliferation. INST-MFA also showed that although NSCs consumed more glucose, they directed a larger portion towards oxidative phosphorylation, whereas astrocytes preferentially metabolized pyruvate to secrete lactate, glutamine, or citrate [110]. 

INST-MFA was also recently used to quantify the metabolic impact of 3D architecture on differentiation of induced pluripotent stem cells (iPSCs) into cardiomyocytes (CMs). iPSC-CMs were cultured in 2D and 3D cell culture conditions, labeled with [1,2-^13^C]Glucose and analyzed using INST-MFA. Flux maps showed that that iPSC-CMs downregulated glycolytic and lipid synthesis fluxes and further unveiled pyruvate metabolism as a distinguishing feature between both groups. Pyruvate was primarily metabolized into lactate in 2D cultures, while 3D cultures directed pyruvate towards the mitochondria. Finally, ATP production was calculated by multiplying the number of ATP molecules in relevant fluxes by respective flux measurements. This analysis showed that ATP production rates were higher in 3D cultures and that a larger portion of ATP was generated from mitochondrial metabolism [111]. 

iMFA was recently used to study whether induced pluripotent stem cells (iPSCs) use extracellular lactate, as well as whether lactate metabolism is modulated by glucose availability. iPSCs were cultured in control, low glucose, high lactate, and high lactate + low glucose conditions. Each group was labeled in parallel with [1,2-^13^C_2_] glucose and [U-^13^C_5_] glutamine. Both high lactate cultures were also labeled with [U-^13^C_3_] lactate. Data from all experiments were integrated into iMFA models. High lactate lowered glucose consumption, lactate secretion, and anaplerotic rates. However, high lactate also promoted malate-aspartate shuttle activity, which transports reducing agents into the mitochondria, as well as fatty acid synthesis. Through iMFA, the authors were able to integrate large, complex datasets into comprehensive quantitative models and show that iPSCs have the metabolic flexibility to incorporate lactate into biosynthetic substrates for proliferation, regardless of glucose availability [112]. Together, these studies show how iMFA can be used to elucidate metabolic differences among differentiating stem cells, which can inform optimal stem cell differentiation and maintenance strategies. 

### 4.2. Cellular Activation

iMFA has been used to elucidate the metabolic activity underlying cellular reprogramming. Monophosphorylate lipid A (MPLA) triggers macrophages to generate antimicrobial molecules but simultaneously disrupts mitochondrial metabolism, forcing cells to rely on glycolysis. Primed macrophages eventually restore mitochondrial metabolism yet continue to be antimicrobial. To understand the underlying metabolic events underlying acute and sustained antimicrobial macrophage activation, macrophages were analyzed via iMFA 24 hours and 72 hours following MPLA treatment. iMFA revealed that untreated macrophages shuttle cytosolic malate into the mitochondria but during acute MPLA treatment, macrophages redirect malate to pyruvate for glycolysis. At 72 hours post-treatment, macrophages further increase glycolytic fluxes but also restore malate shuttling into the mitochondria. Thus, MPLA-treated macrophages temporarily redirect malate, likely to avoid futile shuttling into dysfunctional mitochondria while they upregulate mitochondrial biogenesis and glycolytic metabolism. Sustained glycolytic upregulation is postulated to provide further sources for antimicrobial production. This iMFA study shows the power of this technique to decipher both specific changes in metabolite shuttling and overall metabolic activity to understand how macrophages are reprogrammed by MPLA to fight infection [113]. 

A similar approach was used to study activation of brown adipose tissue, which generates heat through uncoupled respiration and plays a role in whole-body energy expenditure. Previous work demonstrated that cold increases brown adipose tissue glucose and fatty acid uptake as well as glycolysis, β-oxidation, glycogen synthesis, and fatty acid synthesis genes. INST-MFA was used to analyze [U-^13^C] glucose-labeled brown adipose tissue treated with a β3-AR, a β3-adrenergic receptor agonist used to mimic cold exposure. INST-MFA revealed that brown adipose tissue TCA activity increased with agonist treatment as pyruvate contribution to the TCA cycle through pyruvate dehydrogenase (PDH) and pyruvate carboxylase (PC) both increased [114]. A subsequent iMFA study that similarly analyzed murine brown adipocyte metabolism after β3-AR agonist treatment also showed increased both TCA cycle and anaplerotic pyruvate carboxylation fluxes. Mitochondrial pyruvate carrier (MPC) was upregulated in cold temperatures, and its inhibition blocked cold-induced increased TCA cycle flux [115]. 

INST-MFA was also recently used to characterize metabolic differences between resting and activated platelets. Optimal labeling schemes were determined a priori by applying INCA’s software tracer simulation function to a previously curated platelet metabolic model [116]. Simulations revealed that a glucose mixture of 30% [1,2-^13^C_2_]glucose, 50% [U-^13^C_6_]glucose, and 20% unlabeled glucose provide optimal flux resolution for glycolysis/PPP in platelets, while 25% [1-^13^C]acetate, 25% [2-^13^C]acetate, and 50% unlabeled acetate was the ideal acetate mixture to resolve the platelet TCA cycle. Platelets were collected and isolated from whole blood, washed, and incubated with either vehicle control or thrombin. INST-MFA demonstrated that at rest, glucose consumption fuels glycolysis while acetate is the sole source for TCA cycle. Upon thrombin activation, platelets undergo systemic increases in metabolic fluxes and increase glycogen catabolism by 3-fold. Furthermore, INST-MFA showed that acetate consumption remained constant and increased TCA cycle fluxes were the result of redirecting glycolytic flux into the mitochondria. While total PPP flux increased due to increased overall glucose consumption, activated platelets lowered the proportion of glucose that entered the PPP by half. Through INST-MFA, the authors quantified absolute and relative flux changes in platelets for the first time. They also demonstrated that iMFA can be utilized to study metabolism in suspended cells [117]. Together, these examples show how iMFA can be used to identify enzymes that play crucial roles in physiological reprogramming events 

### 4.3. Cancer

iMFA can parse nontrivial cancer cell metabolic reprogramming [1], including changes that occur prior to clinically detectable disease. Human bronchial epithelial cells treated with cigarette smoke condensate increased glucose consumption and lactate production. [U-^13^C_5_] glutamine labeling showed increased glutamine enrichment of acetyl-CoA and [M+5] citrate in treated cells, suggesting a shift towards reductive TCA metabolism of glutamine. iMFA enabled the investigators to determine that cigarette smoke treated cells reversed isocitrate dehydrogenase (IDH) flux from oxidative to reductive carboxylation, a finding which would have been difficult to identify from labeling data alone. iMFA also showed an increased flux of malic enzyme, which contributes NADPH for lipogenesis, as well as increased dependency on de novo lipogenesis rather than exogenous lipid uptake for cellular biomass [118]. 

A similar approach was used to quantify the “Reverse Warburg Effect,” in which cells use lactate as an energy source across normal epithelial, tumorigenic luminal, and metastatic breast cells. High lactate slightly decreased PPP flux in all three cell types but increased citrate production from IDH. iMFA identified significantly increased anaplerotic and TCA fluxes, especially PDH and citrate synthase fluxes, in the metastatic line with no significant changes in the other cell lines. These results suggest that cancer cells modulate mitochondrial metabolism in response to high-lactate environments, such as those in hypoxic tumor regions [119]. 

iMFA is particularly valuable in elucidating alternative or unique metabolic pathways used by cancer cells. For example, lung cancer cells consumed less glucose and glutamine and produced less lactate when they were grown as anchorage-independent spheroids than as anchorage-dependent monolayers [34]. iMFA with traditional reactions and compartmentalization could not fit the spheroid MDV, since citrate demonstrated enhanced reductive carboxylation but palmitate, which is produced from citrate, did not. The iMFA metabolic network was then modified so that cytosolic citrate from reductive carboxylation entered the mitochondria, where it mixed with mitochondrial citrate prior to being used for cytosolic palmitate synthesis. This iMFA model significantly improved the MDV fit. Cytosolic citrate transport into the mitochondria was validated using citrate transporter knockout cells. This work highlights how iMFA can uncover alternative pathways through model iteration, which can then be experimentally confirmed [34]. 

A separate study used iMFA to decipher whether exogeneous citrate is first metabolized in the cytosol or the mitochondria in hypoxic hepatocellular carcinoma (HCCs) cells. In HCCs labeled with [2,4-^13^C_2_] citrate, TCA intermediate patterns were indefinite and showed evidence of direct metabolism in both compartments. The initial tracer analysis interpretation required the simplifying assumption that TCA intermediates were present for only one TCA cycle turn. iMFA, which can account for multiple TCA cycle turns, unveiled significant cytosolic citrate metabolism via ATP-citrate lyase and IDH1, and that exogeneous citrate must be metabolized to α-ketoglutarate before entering the mitochondria. Thus, by taking into account multiple TCA cycle turns, iMFA uncovered compartmentalization of exogenous citrate metabolism [120].

### 4.4. Infection

Effective antibiotic regimens must significantly disrupt bacterial metabolism and overcome the compensatory metabolic strategies employed by resistant strains [121]. Bedaquiline (BDQ) is an antimycobacterial agent that inhibits ATP synthase in *Mycobacterium tuberculosis (Mtb)*. BDQ-treated *Mtb* are believed to upregulate compensatory reactions to restore ATP generation. To directly analyze metabolic reprogramming and identify crucial compensatory metabolic nodes, *Mtb* cells were independently labeled with ^13^C_6_ glucose, ^13^C_2_ acetate, ^13^C_3_ propionate, or NaH^13^CO_3_. Labeling data from all experiments were input into an *Mtb*-specific model that contained the non-mammalian glyoxylate and methylcitrate cycles. iMFA results unveiled increased anaplerotic phosphoenolpyruvate carboxylase (PEPCK) activity and increased flux through the subsequent ATP-generating pyruvate kinase (*PykA*) step. iMFA further confirmed that glucose flux is preferentially rerouted towards the PPP to avoid ATP consumption by phosphofructokinase-1 (*pfkA*). These metabolic nodes were identified as crucial elements in metabolic compensation and confirmed by showing that BDQ abolished *pykA* knockout strains but had no significant effect on *pfkA* strains. This study demonstrates the power of iMFA in identifying bacterial compensatory nodes to facilitate effective antimicrobial therapeutic development [122].

iMFA has also been used to identify and quantify metabolic changes during infection. Viruses hijack host cell metabolism to drive flux towards pathways that are beneficial for viral production [123]. iMFA using [1,2-^13^C] glucose and [U-^13^C] glutamine tracers quantified the metabolic impact of infecting an amniocyte-derived cell line with adenovirus type 5 under both exponential growth and growth arrest conditions. iMFA estimated that cells in exponential growth increased reductive carboxylation and PPP fluxes upon infection. iMFA also demonstrated that cells in both exponential growth and growth arrest convert citrate to cytosolic acetyl-CoA after infection, which contributes to lipogenesis. This further suggests that infection may promote metabolic alterations that support host acetylation, a mechanism that adenoviruses use to control gene expression during infection. Through these studies, iMFA highlighted infection-induced metabolic changes that support adenoviral replication, which is important for therapeutic targeting, as well as strategies to increase virus production for viral and gene therapy vectors [64]. 

### 4.5. Compensating for Genetic Loss of Function

iMFA can shed light on how cells adapt to genetic loss of function and suggest targets for therapeutic development. In one such study, iMFA with [U-^13^C] glutamine and [U-^13^C] glucose was used to evaluate how clinically relevant mitochondrial DNA alterations disrupt central carbon and amino acid metabolism. Cells with mutant mitochondrial DNA decreased oxygen consumption, glucose entry into the TCA cycle, and glutamine-derived citrate. Initial iMFA analysis confirmed that mutant cells increased glucose and glutamine uptake, but the metabolic byproducts were shuttled out of the cell as lactate and glutamate instead of contributing to mitochondrial metabolism. SLC7A11, a glutamine-cystine antiporter, was upregulated in mutant cells. iMFA was used to analyze wild-type cells overexpressing SLC7A11 to explore how this transporter impacted metabolism. iMFA identified enhanced glucose-derived pyruvate flux into the TCA cycle in SLC7A11 overexpressing cells via cystine-mediated production of α-ketobutyrate. Thus, increased cysteine and decreased glutamine metabolism restore glucose oxidation to compensate for mitochondrial DNA mutations [124]. 

iMFA was also used to characterize metabolic reprogramming following deletion of mitochondrial citrate transport protein (CTP), which transports citrate out of the mitochondria for fatty acid synthesis. CTP deficiency leads to neurodevelopmental syndromes, and CTP overexpression is associated with poor cancer prognosis [125]. iMFA of CTP-deficient lung cancer cells with glutamine and glucose tracers showed that CTP deficiency did not eliminate the glucose contribution to lipogenesis due to increased α-ketoglutarate mitochondrial efflux. The cytosolic α-ketoglutarate then underwent reductive carboxylation to citrate via isocitrate dehydrogenase 1 (IDH1) and eventually was used for fatty acid synthesis. iMFA also unveiled two unexpected discoveries. First, glutamine-dependent anaplerosis was reduced but compensated for by glucose-dependent anaplerosis via increased pyruvate carboxylase activity. Second, α-ketoglutarate used for reductive carboxylation was derived from both glucose and glutamine to eventually contribute to lipogenic acetyl-CoA [125].

Finally, iMFA was used to analyze the impact of loss-of-function mutations in fumarate hydratase (FH), a TCA cycle enzyme associated with hereditary leiomyomatosis and renal cell cancer. FH-deficiency is also associated with resistance to respiratory chain inhibitors. Labeling with [1,2-^13^C] glucose and [U-^13^C] glutamine revealed similar glycolytic profiles but slight differences in fumarate and proline labeling between control and cells with diminished FH activity. iMFA unveiled that diminishing fumarate hydratase decreased glucose-derived pyruvate flux into the mitochondria, which lowered TCA cycle activity. Cofactor consumption/generation were included in the iMFA model, and ATP was calculated by using a phosphate-oxygen (P/O) ratio calculation in which each NADH generated 2.3 moles of ATP. This analysis revealed that NADPH consumption and NADH production were reduced in FH-diminished cells, suggesting that the altered redox balance may have caused decreased proline synthesis from glutamine. This analysis also demonstrated that FH-deficiency shifts ATP production towards glycolysis, allowing for increased resistance to mitochondrial inhibitors [126]. Together, these studies exemplify how iMFA can parse key drivers of cancer reprogramming from the multitude of metabolic alterations that occur. 

### 4.6. Drug Effects

Metabolic therapies are being explored for pathologies with significant metabolic perturbations such as cancer and cardiovascular disease. However, the interconnected nature of metabolism requires that we fully elucidate drug effects on systemic metabolism that could diminish their clinical utility. We recently examined how several targeted metabolic therapies designed to inhibit a specific metabolic enzyme in fact impacted diverse aspects of the metabolic network. Human umbilical vein endothelial cells were treated with inhibitors designed to target one of three glycolytic side branch pathways: the hexosamine biosynthetic pathway (azaserine), pentose phosphate pathway (dehydroepiandrosterone; DHEA), or the polyol pathway (fidarestat). Initial tracer analysis showed decreased labeling in TCA and amino acid metabolites in both DHEA and fidarestat groups; however, the full scope of changes was not immediately apparent. Through iMFA, it was revealed that all three groups underwent drastic changes in the TCA cycle, which was theorized to be a result of either compensatory (DHEA, azaserine) or off-target (fidarestat) effects [35].

Some cancerous mutations may actually hinder metabolic flexibility and make the cell more vulnerable to agents that induce metabolic stress. The tumor suppressor liver kinase B1 (LKB1), which senses ATP availability, is mutated in many cancers which may shift the balance between glycolysis and oxidative phosphorylation. To understand the full scope of LKB1-induced metabolic changes, [1,2-^13^C] glucose, [U-^13^C] glucose, and [U-^13^C_16_] palmitate labeling data from LKB1 expressing or non-expressing A549 and H460 human lung cells was integrated into an iMFA model. The results indicated that LKB1 increased mitochondrial metabolic flexibility, enabling the cells to oxidize pyruvate, glutamine, and fatty acids. This change was primarily driven by increased flux through reactions fueling the TCA cycle (e.g., pyruvate dehydrogenase and glutamine anaplerosis) and increased flux through the TCA cycle itself (e.g., α-ketoglutarate dehydrogenase) [127]. LKB1-deficient cells were previously found to be sensitized to the complex I inhibitor phenformin, which induces mitochondrial reprogramming [128]. After iMFA results highlighted the reactions attenuated by LKB1-deficiency, the authors hypothesized that combinatorial treatment of phenformin along with glutaminase inhibitor, BPTES, would improve targeting of LKB1-deficient cells. Cell viability assays confirmed that LKB1-deficient cells are particularly sensitive to the combinatorial treatment and inhibition of glutamine metabolism [127]. 

In another study, MCF-7 human breast carcinoma cells were treated with paclitaxel, a chemotherapeutic that increases microtubule polymerization, after which they were fed [1-^13^C] glucose and [U-^13^C] glutamine. MDVs, specifically alanine labeling, suggested that paclitaxel decreased glycolytic flux while maintaining oxidative PPP flux. iMFA demonstrated that paclitaxel-treated cells increased TCA cycle, malate dehydrogenase, and pyruvate carboxylase pathway activity to compensate for reduced glycolytic flux. ATP levels, which were calculated from the sum of substrate-level phosphorylation events and NADH P/O ratios [129], were elevated in paclitaxel-treated cells, leading the authors to suggest that paclitaxel-treated cells upregulate mitochondrial metabolism to generate energy for ATP-dependent microtubulin polymerization and paclitaxel excretion [130]. 

### 4.7. Extracellular Vesicles

Extracellular vesicles, membrane-bound vesicles that bud off donor cells, may transfer metabolites to recipient cells [131,132]. Exo-MFA is a multicellular modeling approach created to quantify extracellular vesicle metabolite transport fluxes among cells [133]. In the pilot study, extracellular vesicles were collected from cancer-associated fibroblasts (CAFs) labeled with [U^13^-C_6_] glucose or [U^13^-C_5_] glutamine and added to pancreatic ductal adenocarcinoma cell cultures (PDACs). Metabolites from CAFs, PDACs, and extracellular vesicles were collected at each experimental time point and analyzed via GC-MS. The resulting MDVs were fit into a custom iMFA model that included terms for extracellular vesicle cargo packaging, secretion, and cargo release into recipient cells. Exo-MFA was used to calculate the rate of metabolite transfer into recipient PDACs and the intracellular PDAC fluxes. Using this approach, the authors found that TCA metabolic fluxes increased six hours after extracellular vesicle exposure, likely due to lactate, glutamine, and TCA intermediate cargo delivery. This work demonstrates that iMFA can be expanded to estimate quantitative information about multicellular metabolic interactions [133,134].

### 4.8. In Vivo Studies

iMFA has been used to investigate gluconeogenesis in conscious mice in vivo by implanting catheters in the left common carotid artery for tracer infusion and in the jugular vein for blood sampling [32]. In a pilot in vivo study to examine how fasting affects sources for hepatic gluconeogenesis, mice were infused with [6,6-^2^H_2_] glucose, [^2^H_2_] water, and [^13^C_3_] propionate. Flux analysis demonstrated that glycogen contribution to hepatic glucose production diminished with fasting while glycerol and phosphoenolpyruvate contributions increased. The initial iMFA model assumed that CO_2_ was not reincorporated in metabolism, an idea supported by previous in vivo NMR work but contradicted by a GC-MS study [102]. When the model was revised to include CO_2_ reincorporation, the fit improved. Thus, iMFA supported CO_2_ reincorporation into hepatic gluconeogenic metabolism [32]. 

This combined catheterization/iMFA approach was also used to determine how in vivo glucose production changed in hepatic AMP-activated protein kinase (AMPK) knockout mice with treadmill exercise. iMFA showed that AMPK knockout decreased glycogenolysis and hypoglycemia in exercising mice but had a negligible effect on TCA cycle and anaplerotic fluxes [135]. Similarly, the role of hepatic pyruvate carboxylase in hepatic gluconeogenesis was studied in liver-specific pyruvate carboxylase knockout mice provided with a [U-^13^C] glucose infusion. [M+6] glucose was higher in liver-specific pyruvate carboxylase knockout mice, implying decreased hepatic gluconeogenesis. An iMFA model revealed that whole body gluconeogenesis was maintained in liver-specific pyruvate carboxylase knockout mice, suggesting an extrahepatic glucose source [136]. 

In vivo iMFA was also used to study how diet may impact nonalcoholic steatohepatitis (NASH) progression, which is believed to result from oxidative stress [137]. Mice were fed either normal chow or Western diet along with the antioxidant Vitamin E for 8 or 20 weeks. Vitamin E failed to reduce liver damage and actually increased signs of metabolic dysfunction such as whole-body adiposity, reduced lean mass, and increased blood glucose in chow-fed mice but not Western diet fed mice. iMFA revealed that the specific metabolic perturbations with Vitamin E for mice on the two diets were different. Vitamin E in normal chow mice lowered TCA flux, suggesting cofactor reduction, lowered ATP demand, or lower oxidative respiration. Vitamin E in Western diet mice increased TCA flux, pyruvate flux, and glucose production at 8 weeks but had no effect on metabolic fluxes after 20 weeks [137]. These results indicate that Vitamin E may exacerbate insulin resistance effects and increase reliance on the TCA cycle for cofactor production.

## 5. New Frontiers in iMFA 

### 5.1. Dynamic MFA

While non-stationary iMFA can account for systems that are not at isotopic steady state, both standard and non-stationary iMFA require that the system is at a metabolic steady state. However, this assumption cannot be met for every system, especially for cases in which temporal flux changes are of interest. Mammalian cells change their metabolic activity over time, especially during differentiation, activation, and proliferation [138]. Dynamic ^13^C-MFA (DMFA) is an emerging non-steady state approach; however, DMFA faces unique experimental and computational challenges, since metabolite concentrations, MDVs and fluxes that vary with time greatly complicate flux estimation [25]. One DMFA approach divides the experiment into time intervals and performs iMFA on each phase, creating a time profile of flux change [139,140]. These studies represent important advancements; however, current isotope-assisted DMFA primarily focus on microorganisms [141], and the limited mammalian DMFA studies use external flux data only without incorporating intracellular isotope metabolite labeling data [142]. Further work is needed to determine if DMFA can be used to analyze complicated isotope patterns from mammalian cells.

A recent proof-of-concept study used B-spline curve fitting to estimate non-steady state glycolytic fluxes from isotopomer data acquired from adipocytes treated with insulin and labeled with ^13^C_6_ glucose [142]. The authors sought to develop a feasible DMFA approach that overcame previous limitations such as infeasible solutions (e.g., negative isotopomer and metabolite abundances) and poor algorithmic stability. Metabolites were sampled across six time points ranging from 1-60 min. This approach demonstrated that following insulin treatment, glycolytic pathways responded faster and reached a higher flux magnitude than the TCA cycle. The adipocytes therefore converted the majority of insulin-dependent glucose influx to lactate. Despite reproducing trends from the experimental dataset, the model did not achieve a statistically acceptable fit, possibly due to underestimating measurement uncertainty. Nevertheless, this work provides a significant advancement in DMFA workflow to resolve non-stable fluxes in labeled human cells and highlights the exciting potential for this approach. 

### 5.2. Genome-Scale MFA

iMFA typically focuses on a subset of metabolic pathways, with most studies described in this review focused on glycolysis and TCA and sometimes the PPP. However, exclusion of pathways deemed to be non-essential or difficult to measure can bias the results, since metabolic pathways are highly interconnected. Genome scale models comprehensively represent all metabolites and reactions in the biological system to offer a true systemic perspective. Several factors limit iMFA model expansion to the genome scale, including lack of information on atom transitions and compartmentalization, as well as experimental design challenges [143]. One study constructed a model with 697 reactions and 595 metabolites to perform ^13^C iMFA in *E. coli* at steady state [144]. Flux ranges for selected reactions (especially in the TCA cycle) expanded due to the possibility of flux through alternate pathways, indicating that genome-scale models will require additional experimental data to resolve peripheral fluxes. 

This model was modified to study transient metabolism in the model cyanobacterium *Synechocystis* 6803 (729 reactions and 679 metabolites) and *Synechococcuss* 2973 (700 reactions and 667 metabolites) using INST-MFA [38,145]. Genome-scale MFA resulted in significantly altered flux distributions from the core model and improved fit to the experimental data in both studies [38,145]. Although these models are much smaller than a human genome-scale model, which contains over 10,600 reactions and 5,600 metabolites, the model size is much larger than any mammalian study iMFA model. A recent study developed a novel parallelized algorithm to perform INST-MFA at genome scale with improved speed [146]. This approach split the network into connected subcomponents and ran simultaneous computation on the networks in parallel. Three models were tested, including the previously mentioned *Synechococcus elongatus* 2973 model, resulting in up to 15-fold acceleration in computation time. Other studies use an alternate approach known as known as two-scale MFA, which uses data from iMFA to constrain flux predictions in a genome-scale model [143,147,148,149].

One possible issue with genome-scale models is that large metabolic networks with limited experimental data can produce large flux variabilities. A potential solution is parsimonious ^13^C MFA, which runs a second optimization after standard iMFA to find the solution that minimizes the sum of all the reaction fluxes [150]. Parsimonious iMFA is based on the principle that evolutionary pressure is likely to select for minimal energy expenditure and therefore selects the flux solution that minimizes the weighted sum of all fluxes. This method also addresses unrealistic solutions that predict large fluxes through futile cycles which are not realistic in vivo. Parsimonious iMFA can also be used in combination with transcriptomic data to minimize fluxes associated with low gene expression, offering another method to constrain fluxes with limited MDV measurements. A recent proof-of-concept study demonstrates that this approach improves flux predictions compared to traditional iMFA; however, its applicability to genome-scale models is limited by its computational complexity [150].

### 5.3. Co-Culture and Cross-Talk

Co-culture studies are vital for physiologically relevant model systems, since different cell types are in constant metabolic communication with one another. For example, blood–brain barrier models best recapitulate barrier integrity and function when endothelial cells are cultured alongside other brain cells such as astrocytes and neurons [151]. Experimental methods to analyze metabolism in co-cultures require physical separation of distinct cell populations, such as purification or fluorescent sorting, which risk disrupting metabolic measurements. The potential for iMFA to analyze fluxes in co-culture systems without separating cells experimentally was demonstrated in a proof-of-concept study [152]. Cells were modeled as separate compartments, each with a complete set of reactions and an “f1” parameter that represented the population fraction of species 1. Knowledge of species-specific differences, such as preferences at key metabolic pathways, was also required. Approach feasibility was demonstrated in an *E. coli* co-culture system, in which one strain was an upper glycolysis knockout and the other strain was a PPP knockout [152]. The model was also extended to secreted metabolites that were exchanged between the two systems. This approach may empower studies focused on quantifying metabolic interactions and transport among cell types, especially at endothelial or epithelial interfaces. However, further work is required to identify optimal tracer selection and develop a priori knowledge of distinct metabolic differences between co-cultured eukaryotic cell types before it can be readily deployed for mammalian co-culture studies. 

## 6. Conclusions

The advancement of mathematical and computational frameworks has enabled iMFA to be applied to complex, compartmentalized mammalian cells, facilitating studies focused on human physiology and disease. The strength of iMFA lies in its ability to generate flux maps from complex metabolomic data, allowing end users to quantify metabolic differences among cell states. iMFA can highlight systemic metabolic shifts in multiple pathways yet also assist in identifying crucial nodes and specific enzymatic targets. Furthermore, iMFA can be a powerful tool in generating new hypotheses or complementing other -omics techniques. iMFA has been especially useful in resolving TCA or mitochondrial features such as pyruvate transport dynamics, upregulated redox-associated fluxes, unconventional shuttles, and alternative metabolic programming events. Many of these findings cannot be intuited from isotope labeling or smaller-scale fluxomic methods, owing to the complexity of metabolism. 

We envision that iMFA will continue to yield new insights into metabolic states in human health and disease, especially as larger models and dynamic approaches are developed. Most current in vitro iMFA applications are in cancer while in vivo studies have only examined hepatic gluconeogenesis thus far. However, iMFA has great potential in iPSC differentiation, rare inherited metabolic disorders, and diseases related to metabolic dysfunction such as cardiovascular disease, diabetes, and obesity. As iMFA software and tutorials continue to develop, iMFA approaches should be increasingly accessible for biomedical researchers and lead to new insights into metabolic mechanisms in disease.

## Figures and Tables

**Figure 1 metabolites-12-01066-f001:**
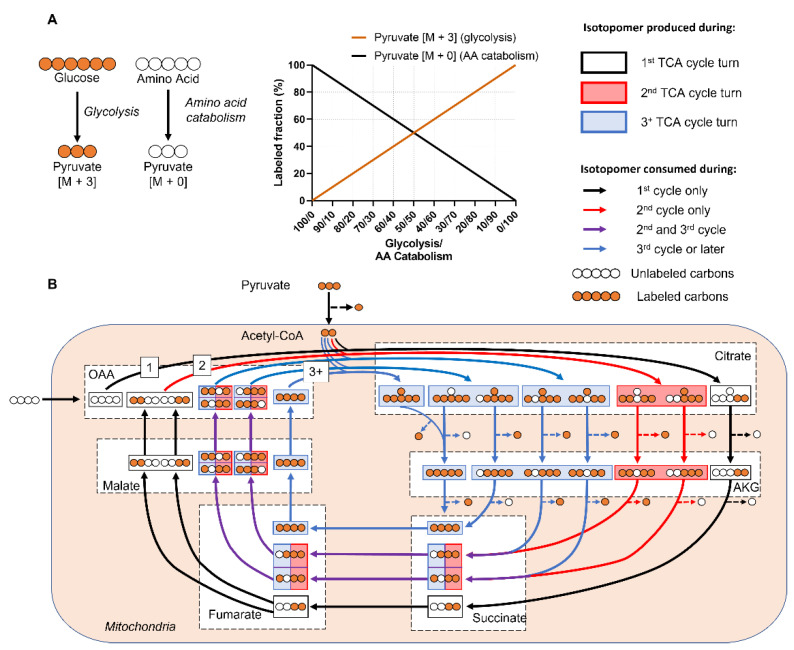
Labeling patterns from mammalian metabolic networks are complex and difficult to interpret. (**A**) A simple metabolic network that assumes only two sources for pyruvate: labeled glucose and unlabeled amino acids. The relative fraction of [M+3]/[M+0] pyruvate isotopomers depends on relative activity of glycolysis vs. amino acid catabolism. As glycolytic activity increases, more of the pyruvate pool will be labeled. (**B**) Cyclic metabolic networks, such as the TCA cycle, generate a different set of isotopomers at each turn. This simplified example assumes a TCA cycle in which uniformly labeled glucose and an unlabeled 4-carbon substrate that feeds into oxaloacetate (OAA) are the only carbon contributors. Each color represents a different turn of the TCA cycle. The anaplerotic source forms the initial OAA backbone that is metabolized with glucose-derived labeled acetyl-CoA to form [M+2] citrate (black arrows). Fumarate and succinate are symmetrical molecules, means the labeled atoms can flip positions. This doubles the number of malate isotopomers generated from partially labeled fumarate (purple arrows) starting from the 2nd turn of the TCA cycle, which ultimately results in additional citrate isotopomers in the subsequent turns of the TCA cycle (blue arrows). Real world examples in which multiple anaplerotic pathways feed into the TCA cycle, as well as reversible reactions, add more complexities. Abbreviations: Citrate (Cit), Alpha-ketoglutarate (AKG), Oxaloacetate (OAA), Fructose-6-phosphate (F6P), Glucose-6-phosphate (G6P), Amino Acid (AA).

**Figure 2 metabolites-12-01066-f002:**
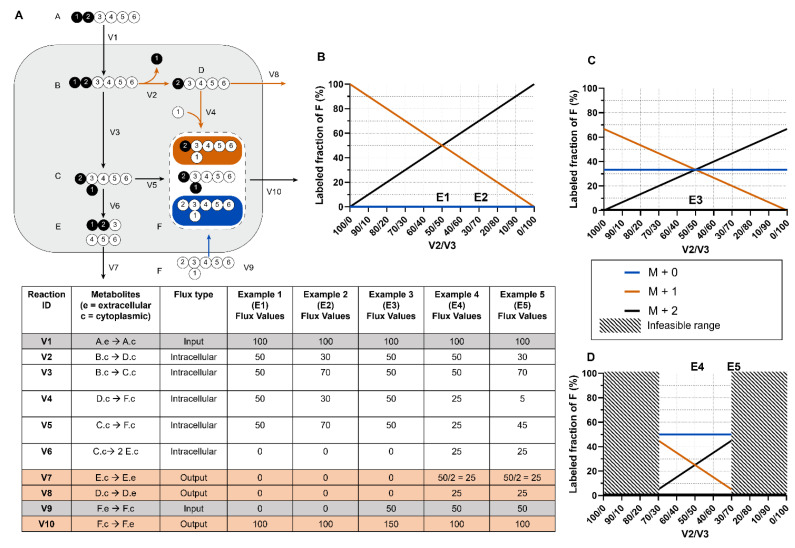
Metabolite MDVs are flux-weighted averages of precursor substrate MDVs. (**A**) A toy model of a metabolic system with two input fluxes, three output fluxes, and five irreversible forward reactions. The corresponding reaction information and flux values for examples are shown in the table below. Note that the measured flux of V7 must be divided by 2 when calculating the amount of C consumed to properly account for reaction stoichiometry. (**B**) The relationship between V2/V3 flux values and metabolite F isotopomers in the scenario where V1 and V10 are the only non-zero fluxes. In these examples, [M+1] and [M+2] are the only possible isotopomers of F. (**C**) The relationship between V2/V3 flux values and metabolite F isotopomers when V9 is non-zero. Possible isotopomers [M+0], [M+1], and [M+2] due to influx of unlabeled F. The influx of unlabeled F dilutes the labeled pool of [M+1] and [M+2]. (**D**) Scenario where V7 and V8 fluxes are both non-zero, which also mandates non-zero fluxes in V2 and V3. This further narrows the range of possible flux values for V2 and V3.

**Figure 3 metabolites-12-01066-f003:**
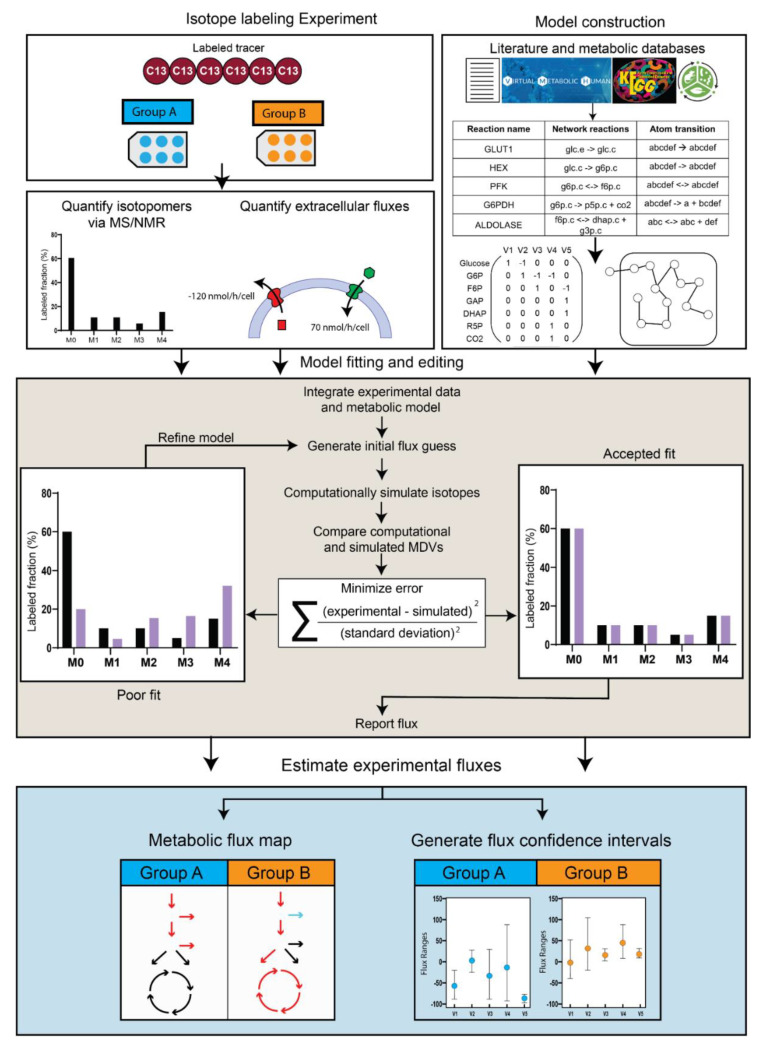
Isotope-assisted Metabolic flux analysis workflow. The three major inputs for iMFA are isotope labeling experiment data, extracellular fluxes, and a metabolic model. The model can be constructed using literature and publicly available databases that delineate the metabolic reactions and atom transitions specific to the organism, tissue, or cell of interest. For most software, this network can be input as a table of reactions along with the corresponding stoichiometry and atom transitions. The table gets converted to a mathematical representation within the user-friendly software. Once the experimental data is fitted to the reaction model, the software generates an initial flux estimate and uses this to computationally simulate MDVs. The fit between simulated and experimentally derived MDVs is assessed, and the process iterates by refining the flux values until a satisfactory fit is achieved. The final flux estimates can be used to output a quantitative metabolic flux map and generate confidence intervals for each flux in the network.

**Figure 4 metabolites-12-01066-f004:**
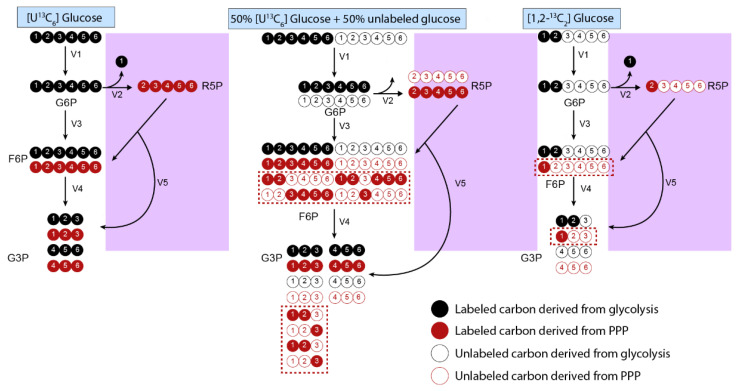
Tracer compositions should be scrutinized to optimize information for pathways of interest. The example displays a simple glycolytic pathway in which we assume that reactions only flow in the forward direction. Uniformly labeled glucose results in identical isotopomers from the PPP (purple box) and glycolysis. Using a combination of 50% fully labeled glucose with 50% unlabeled glucose can better inform PPP fluxes, as the complex atom rearrangement in the PPP leads to M+1, M+2, M+4, and M+5 isotopomers (shown in red box). The percent composition of these isotopomers can be used to determine flux activity at the PPP branchpoints. A more direct option is to use [1,2-^13^C_2_] glucose, which generates an M+1 isotopomer after entering the PPP by being metabolizing into R5P that is further metabolized into M+1 F6P and G3P. In contrast, the glucose that is directly metabolized into F6P generates M+2 isotopomers. Therefore, the ratio of M+1 and M+2 isotopomers in metabolites such as F6P and G3P, as well as derivatives such as lactate and pyruvate, can be compared to determine relative activity at the PPP branchpoint. Red = carbons that pass through the PPP; black = carbons that pass through the main glycolytic pathway. The red, dashed boxed encompasses isotopomers that can only be formed after passing through the PPP. Abbreviations: glucose-6-phosphate (G6P), ribose-5-phosphate (R5P), fructose-6-phosphate (F6P), glyceraldehyde-3-phosphate (G3P), pentose phosphate pathway (PPP).

**Figure 5 metabolites-12-01066-f005:**
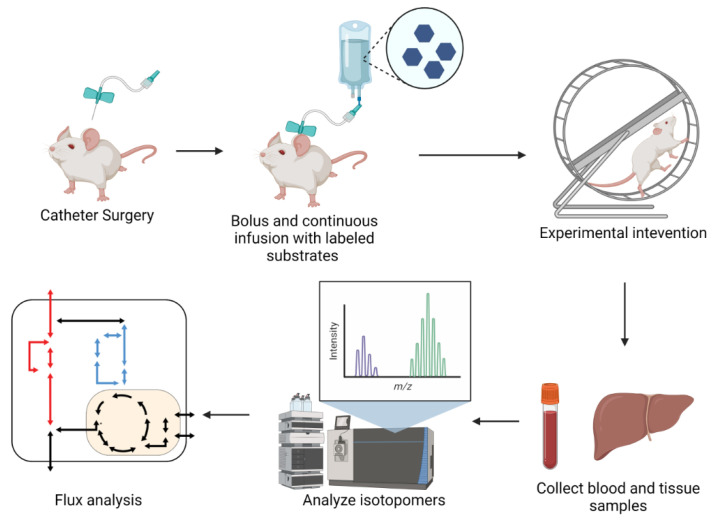
Schematic of in vivo experimental procedures. Schematic of typical in vivo iMFA experimental workflow. Protocols typically begin with catheterization surgeries. Following recovery, mice are provided with bolus of stable isotope substrates and subjected to experimental procedure until metabolites achieve isotopic steady state. Blood and tissue samples are collected for analysis with NMR or MS. Isotopomer data is regressed with iMFA software to quantify metabolic fluxes. Figure created with BioRender.

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
