# Peer review of "Isotope-Assisted Metabolic Flux Analysis: A Powerful Technique to Gain New Insights into the Human Metabolome in Health and Disease"

_metabolites, 2022, doi:10.3390/metabo12111066_

Round 1

Reviewer 1 Report

Overall, this is a thorough review of MFA applied to biomedical systems. There are several small edits suggested below.

Somewhere early in the review, the authors should define isotopic and metabolic steady state for the reader. I think they also should mention an implicit assumption of uniform environmental conditions and homogeneity of cells. That is there are no micro-environments or cells of different lineage affecting the fluxes.

Line 72, the number of measurable isotopomers increases with more sophisticated measurement techniques, not the actual number of isotopomers.

Line 75 "as" demonstrated

Lines 357-359 are confusing, not relevant and can be deleted. Why are plants single nutrient organisms? If so, shouldn't it be CO2 instead of glucose?

In section 3.7 should metabolic channelling also be considered as a factor that leads to some pools being highly unlabeled than expected?

Line 369 "superior"

Line 801- were

Line 976- why include reference 140?  I don't understand how focus on microorganisms is problematic. It isn't relevant to this review, but microorganisms have cyclic and parallel metabolic pathways as well.

Reviewer 2 Report

Summary:

In this review article, the authors provided a wide-scope discussion about MFA and its application in human research. The strength of this article is the comprehensive discussion about MFA’s application on human related research, which is not frequently seen in other MFA reviews. However, certain issues including the use of terminology and some careless writing errors need to be addressed:

Major comments:
P2L64-L75: The concept of “isotopomer” was misused in this manuscript. “isotopomer” refers to isomers having the same number of each isotope of each element but differing in their positions. The more appropriate word here might be “isotopologue”, which refers to molecules that differ only in their isotopic composition (e.g. m+0, m+1, m+2 etc.) or the more general term “isotopic isomer”. It should be noted that MS can distinguish isotopologues but cannot distinguish isotopomer.

P2L88: the term of “iMFA” may cause confusion. The well accepted concept of MFA already assumes the use of isotope tracers. The flux analysis method that does not use isotope tracers (e.g. stoichiometric flux analysis and flux balance analysis) are normally not considered as MFA. Is there any difference between iMFA and MFA? If so, the author need to explain it explicitly.

P3L109: “Fumarate is a symmetrical molecule, means the labeled atoms can flip positions, doubling the number of isotopomers generated from partially labeled succinate.” Actually, the symmetrical effects starts from succinate as succinate is already a symmetric molecule.

P6L229: For cultured cell, the external fluxes can be determined as metabolite accumulation / consumption rate. However, in an in vivo system, the concentration of metabolites remain constant at steady state. Since this review also focus on human clinical experiments, the author should also briefly mention in this section how the external fluxes (e.g. production and consumption of glucose) are determined in an in vivo system.

P10L352-363: The content of this paragraph is only partially correct. The main purpose of tracer design is to differentiate the labeling pattern of substrates of the converging fluxes. For example, fructose-1,6 bisphosphate (FBP) could be made from either F6P or DHAP+GAP. 100% U-13C cause the product of the two pathway exactly the same labeling pattern and therefore not suitable. The reason 50% U-13C glucose is better than 100% U-13C is because with 50% U-13C glucose, GAP+DHAP can generate a differently labeled FBP than F6P does. Therefore, a good tracer strategy is not about how much percent the tracer is applied but to ensure the labeling pattern of substrates of the converging fluxes are different. For example, an even better strategy than 50% U-13C glucose is to use 100% 1,2-13C2 glucose. This tracer is better because it not only have the benefit of 50% U-13C glucose, but also distinguished the labeling pattern of GAP and DHAP. See reference 24 for detailed explanation.

P13L531-533: The example here is not very typical since most of the modern instruments have sufficient resolution to distinguish 13C from 15N. In fact, more often the correction was done for the endogenous 13C, 18O and the impurity of tracer. See (PMID: 28471646) and (PMID: 34193963) for detailed explanation on this topic.

Minor comments:
P1L23: keywords are a little bit redundant here. For example, the concept of 13CMFA is already covered by isotope-assisted metabolic flux analysis.

P2L75: “as demonstrated recently” instead of “a demonstrated recently”

P2L76: “to analyze” instead of “to analyzing”

P3L113: There is legend for Figure 1C but no actual figure.

P4L129: should use bolt for (A), (B), (C) etc.

P4L142: secretion fluxes (V7, V8, and V10) instead of (V7, V8, and V9).

P7L257-259: The paragraph strangely breaks into two small paragraphs.

P11L407: 13C-MFA refer to any MFA that used 13C, not necessarily require steady state.

P15L612-622: The paragraph strangely breaks into two small paragraphs.
